# Process Analytical Technology Obtained Metastable Zone Width, Nucleation Rate and Solubility of Paracetamol in Isopropanol—Theoretical Analysis

**DOI:** 10.3390/ph18030314

**Published:** 2025-02-24

**Authors:** Mahmoud Ranjbar, Mayank Vashishtha, Gavin Walker, K. Vasanth Kumar

**Affiliations:** 1Synthesis and Solid-State Pharmaceutical Centre, Department of Chemical Sciences, Bernal Research Institute, University of Limerick, V94 T9PX Limerick, Ireland; 2Chemical and Process Engineering, Faculty of Engineering and Physical Sciences, University of Surrey, Guildford GU2 7XH, UK

**Keywords:** crystallization, kinetics, metastable zone width, PAT, thermodynamics, nucleation

## Abstract

**Background:** Metastable zone width (MSZW) and solubility are crucial for developing crystallization procedures in the purification of active pharmaceutical ingredients (APIs). Traditionally, determining these properties involves labor-intensive methods that can take weeks or even months. With advancements in process analytical technologies (PAT) and the increasing focus on quality by design (QbD) in pharmaceutical manufacturing, more efficient and reliable protocols are needed. In this study, we employ in situ Fourier Transform Infrared (FTIR) spectroscopy and Focused Beam Reflectance Measurement (FBRM) to establish protocols for measuring solubility at different temperatures and MSZW at varying cooling rates. **Methods:** We experimentally determined MSZW and solubility using FTIR spectroscopy and FBRM. IR spectra were analyzed to obtain solubility concentrations, while FBRM counts were used to extract MSZW and supersolubility concentrations. The collected data were assessed using four theoretical models, including a newly developed model based on classical nucleation theory. By fitting experimental MSZW data to these models, we determined nucleation kinetics and thermodynamic parameters. **Results:** Our novel model exhibited excellent agreement with experimental MSZW data across different cooling rates, demonstrating its robustness. The nucleation rate constant and nucleation rate ranged between 10²¹ and 10²² molecules/m³·s. The Gibbs free energy of nucleation was calculated as 3.6 kJ/mol, with surface energy values between 2.6 and 8.8 mJ/m². The estimated critical nucleus radius was in the order of 10⁻³ m. **Conclusions:** The protocols we developed for predicting MSZW and solubility of paracetamol using PAT can serve as a guideline for other APIs. Our theoretical model enhances the predictive accuracy of nucleation kinetics and thermodynamics, contributing to optimized crystallization processes.

## 1. Introduction

Crystallization is an important unit operation across various industries, notably in pharmaceuticals, where it serves as a primary method for purifying active pharmaceutical ingredients (APIs) [1,2,3,4,5] Approximately 90% of APIs and 70% of solid chemicals achieve their purest forms through crystallization processes [6,7]. This process is fundamentally driven by supersaturation, which serves as the driving force for nucleation and crystal growth [5]. Industries exploit crystallization to obtain products with specific properties, including crystal form, size and purity [8]. While nucleation can enhance the purity of crystalline products, it often leads to heterogeneity in terms of purity and results in crystals with a wide size distribution. In some cases, nucleation can produce agglomerated crystals, adversely affecting downstream unit operations. Uncontrolled agglomeration during crystallization can negatively impact product quality, including purity and crystal size distribution. In extreme cases, nucleation can lead to the formation of undesirable polymorphs, which may have different physical properties and stability profiles [9,10,11]. In such scenarios, controlled crystal growth becomes the preferred method for API purification. The crystal growth process is inherently influenced by the quality of the seed crystals; therefore, product attributes such as polymorph type and crystal size distribution can be controlled through the careful selection and preparation of these seeds [12]. Understanding nucleation and growth theories is essential; however, precise experimental data on solubility and the metastable zone width (MSZW) are crucial for identifying optimal growth conditions and developing standard operating procedures (SOPs) for crystallization experiments [5,13].

The MSZW represents the range of supersaturation within which a solution remains metastable before crystallization occurs [14]. This zone is influenced by factors such as cooling rate, solution history, agitation, solution volume, and vessel geometry [15,16,17,18,19,20,21,22]. Accurate knowledge and data on the MSZW guide experimentalists and process engineers in designing crystallization processes targeting primary nucleation, secondary nucleation or controlled crystal growth. In this study, we present solubility and metastable zone width (MSZW) data for paracetamol (acetaminophen) in isopropanol. Paracetamol is frequently utilized as a model compound in crystallization research to develop new processes and standard operating procedures (SOPs) [23,24,25,26,27,28,29]. For instance, studies have employed paracetamol to investigate sonocrystallization techniques and to develop protocols for continuous manufacturing of active pharmaceutical ingredients (APIs) [27,30]. Nagy et al. demonstrated the utility of paracetamol in continuous crystallization setups involving milling units positioned either upstream or downstream in the process [31]. The solubility and MSZW data we obtained over a wide temperature range for paracetamol in isopropanol are valuable to the crystallization community.

We developed SOPs utilizing state-of-the-art process analytical technology (PAT) tools, including in situ Fourier Transform Infrared (FTIR) spectroscopy and in situ Focused Beam Reflectance Measurement (FBRM). This approach enabled us to obtain high-quality solubility and MSZW data, adhering to Good Manufacturing Practice (GMP) standards and the Quality by Design (QbD) approach in pharmaceutical manufacturing. The PAT-assisted SOPs developed in this work enable the acquisition of solubility and MSZW information across various temperatures in less than 24 h, a significant improvement over conventional methods that can take weeks or months. We also present the mathematical approach that we used to convert the PAT-obtained IR spectra and FBRM counts into solubility concentration, MSZW or supersolubility concentration.

Finally, to complement the experimental results, we analysed the MSZW data using theoretical models to predict nucleation parameters that include the nucleation kinetic constant, order of nucleation and the nucleation rate as a function of the starting solubility temperature. These models include the ones developed by Nyvlt [32], Sangwal [33] and Kubota [34]. Furthermore, we propose a new theoretical model based on classical nucleation theory to predict nucleation rates at different cooling rates, addressing the limitation of existing models that assume nucleation rates are independent of cooling rates. Our model provides insights into both kinetic and thermodynamic nucleation parameters, such as nucleation rate, Gibbs free energy of nucleation, surface energy (interfacial tension), critical nucleus size and the approximate number of unit cells in the critical nucleus. More importantly, we demonstrate that the newly derived mathematical expression enables the calculation of nucleation parameters across different cooling rates, offering a significant advancement in crystallization modelling.

## 2. Results and Discussion

### 2.1. PAT-Obtained Solubility

In Figure 1a, we show the plot of FTIR intensity obtained at 1516 cm^−1^ as a function of the solution temperature obtained using Recipe 1. In Figure 1a, we also show the initial intensity and final intensity together with some information about the temperature at which all the solids become completely dissolved. From Figure 1a, it is evident that all the solids in solution dissolved completely when the solution temperature was equal to 55.59 °C, as the IR intensity increases exponentially up to this temperature. This means the solubility of paracetamol in isopropanol at this temperature is equal to the mass of the solids added to the reactor in the 70 mL of isopropanol (IPA), or 185.71 g of paracetamol per litre of isopropanol. Another noteworthy observation is that, at all the temperatures above 55.59 °C, the IR intensity decreases linearly with further increases in temperature up to 65 °C. This clearly exposes the experimentally observed effect of temperature on the IR intensity. The IR intensity decreases with an increase in temperature. To cancel the effect of temperature, we calculated the slope from the plot of IR intensity versus temperature in the range of 55.50 °C to 65 °C (see the linear trend line observed at *T** > 55.5 °C in Figure 1a). The slope was then subtracted from the raw IR data to cancel the temperature effects. To predict the solubility calculations, we used this processed data wherein we subtracted the temperature effects from the raw data obtained directly from the PAT.

The temperature effect-corrected FTIR intensity was then converted into the solubility concentration *C** using Equation (46), discussed in Section 3.1. The solubility concentration, calculated based on the FTIR intensity, was plotted against the solution temperature, as shown in Figure 1b. The *C** increased exponentially with *T**. The relationship between the solubility of paracetamol obtained using the in situ FTIR versus temperature follows the following relationship:*C**= 0.0071*T**^2^ + 1.9797*T** + 56.318(1)
where *C** is the solubility concentration in g/L and *T** is the solubility temperature in °C. The solubility data obtained from our studies deviate from the ones reported in the literature (see Figure 1b). To ensure our values were correct, we repeated the solubility experiments thrice, following three different protocols (see Recipe 1, Recipe 2 and Recipe 3 in Section 2.1). Firstly, we calculated the solubility using Recipe 1, described in Section 3.1, where the solubility was obtained at a heating rate of 0.05 K/min, starting from a temperature of −2 °C. Then, the second solubility experiment was repeated at a slightly slower heating rate of 0.01 K/min, starting from a temperature of −8 °C, following Recipe 2, discussed in Section 3.1. In Figure 1b, we show the plot of the solubility concentration *C** versus the solution temperature *T** obtained using Recipe 1 and Recipe 2. From Figure 1b, irrespective of the heating rate used to measure the solubility, the solubility of paracetamol remains the same, and plots almost overlap each other. This means the solubility values predicted using both the recipes overlap each other and produce the same results. Finally, the solubility was obtained a third time using Recipe 3 to ensure that the heating rates used in the first two solubility experiments (i.e., in Recipe 1 and Recipe 2) were sufficient to dissolve the paracetamol to a concentration that equals the solubility of curcumin at the range of studied temperatures (−2 °C to 65 °C for Recipe 1 and −8 °C to 65 °C for Recipe 2) described in these recipes. As mentioned in Section 3.1, this was achieved by executing Recipe 3, which involves measuring IR intensity from 30 to 35 °C at a heating rate of 0.05 K/min followed by a waiting time of 120 min at 35 °C. Furthermore, in this experiment we also measured the IR intensity from 50 to 55 °C at a slow heating rate of 0.05 K/min followed by a waiting time of 120 min at 55 °C. This protocol or recipe was developed to ensure that the first two solubility experiments, performed at heating rates of 0.05 K/min and 0.01 K, provide a sufficient amount of residence time for the paracetamol to dissolve to a concentration that is equal to the solubility concentration at the studied temperatures. If the heating rate is not sufficient to dissolve paracetamol that is equal to *C** at 35 °C and 55 °C, then it should be reflected in the observed FTIR signals by an increase in the FTIR intensities at these two temperatures during the waiting period of 120 min. However, if the selected heating rate is sufficient to achieve the solubility concentration at the reactor temperature (e.g., 35 °C), then the FTIR intensity should remain constant, as it becomes impossible to dissolve more than *C^*^* in the solution even if we maintain the reactor temperature at 35 °C for an infinite amount of time. In Figure 1c, we show the plot of the temperature and IR intensity as a function of time. It is evident from Figure 1c that the IR intensity remains constant (equal to 0.1995) when the solution temperature is at 35 °C for the entire waiting time of 120 min. Likewise, as shown in Figure 1c, we also noticed that the FTIR intensity remained constant starting from a temperature of 54.35 °C, and it remained constant (equal to 0.2468) for the solution temperatures above and equal to 54.35 °C and at 55 °C during the waiting period of 120 min. This, essentially, indicates that the initial mass of paracetamol added to the reactor when time *t* = 0 was completely dissolved at 54.35 °C. In other words, the solubility concentration of paracetamol at 54.35 °C is equal to 12.769 g/kg of solvent. More importantly, the solubility concentration at 54.35 °C matches the solubility obtained using the first two recipes. This clearly indicates that the selected heating rate of 0.05 K/min is experimentally valid to perform the solubility experiments using the protocols discussed as Recipe 1 and Recipe 2 in Section 3.1.

The solubility data measured using the in situ FTIR was fitted in the van’t Hoff’s expression, given by(2)$$\ln x=-\frac{∆H_{s}}{RT^{*}}+\frac{∆S_{s}}{R}$$
where *x* is the mol fraction solubility of paracetamol in isopropanol, *R* is the gas constant expressed in terms of J/mol.K, *T* is the temperature in K, ∆*H*_s_ is the enthalpy of solution and ∆*S*_s_ is the entropy of the solution. Theoretically, a plot of *ln x* versus 1/*T** should be linear. In Figure 1d, we show the plot of ln *x* versus 1/*T** for the case of paracetamol in isopropanol. The coefficient of determination *r^2^* between the calculated ln *x* versus 1/*T* and the best-fit trend line was found to be equal to 0.9943; clearly, there exists a perfect linear relationship between these parameters. The heat of solution required to estimate the nucleation rate using the Sangwal model (discussed later) is obtained from the slope using Equation (2) and was found to be equal to 16.28 kJ/mol. Likewise, we obtained the entropy from the intercept using Equation (2), and it was found to be equal to 30.45 J/mol.K.

### 2.2. PAT-Obtained Metastable Zone Width

In Figure 2, we show the plot of solubility versus solubility temperature together with the supersolubility concentration obtained at different $T_{ref}^{*}$. The supersolubility concentration was obtained using a polythermal method [36] which is discussed in detail in Section 3.2. For demonstration purposes, in Figure 2 we also show the plot of the number of counts measured using FBRM during the measurement of MSZW as a function $T_{ref}^{*}$ at a cooling rate of 0.01 K/min. For the sake of convenience and to better visualize the data, we present only the normalized particle counts (so that the counts range from 0 to 1) in Figure 2. The distance between the temperature at which we observe a phase change, $T_{nuc}^{*}$, and the corresponding $T_{ref}^{*}$ represents the ∆*T_max_*, or the MSZW (also see Section 3.2 and Figure 2) at $T_{ref}^{*}$. As shown in Figure 2, the difference between the solubility concentration at $T_{ref}^{*}$ and the solubility concentration at $T_{nuc}^{*}$ (i.e., at the point of nucleation) is equal to the ∆*C_max_*. In Figure 2, the plot of the solubility concentration at $T_{ref}^{*}$ versus the corresponding $T_{nuc}^{*}$ defines the supersolubility line of the paracetamol. It should be noted that at any point on the solubility line, the corresponding ordinate and abscissa is equal to the supersolubility concentration at $T_{ref}^{*}$ and the $T_{nuc}^{*}$ observed experimentally with respect to the $T_{ref}^{*}$, respectively. As expected, the supersolubility concentration is not well defined as the solubility line, and it varies with the cooling rate. The higher the cooling rate, the higher the supersolubility concentration is with respect to the $T_{ref}^{*}$. A noteworthy observation is that the higher cooling rates lead to elevated supersolubility concentrations at corresponding nucleation temperatures compared with those observed at lower cooling rates.

The relationship between the supersolubility concentration *c_s_* versus the nucleation temperature (i.e., $T_{nuc}^{*}$) follows the following relationships:(3)$$c_{s}=81.64e^{0.0162T_{nuc}^{*2}};\;\mathrm{for}dT/dt=0.01\;K/\min$$
(4)$$c_{s}=84.828e^{0.0167T_{nuc}^{*2}};\;\mathrm{for}dT/dt=0.05\;K/\min$$

Note that Equations (3) and (4) are valid only if the $T_{nuc}^{*}$ is expressed in °C.

Likewise, the relationship between the MSZW and the $T_{ref}^{*}\;$(expressed in terms of °C) follows these mathematical expressions:(5)$$\Delta T_{\max}=0.0074T_{ref}^{*2}-0.6692T_{ref}^{*}+18.816;\;\mathrm{for}dT/dt=0.01\;K/\min$$(6)$$\Delta T_{\max}=0.0072T_{ref}^{*2}-0.6193T_{ref}^{*}+20.521;\;\mathrm{for}dT/dt=0.05\;K/\min$$

As detailed in the next section, the relationship between the metastable zone width (MSZW) and the reference solubility temperature ($T_{ref}^{*}$), or between the supersolubility concentration and the nucleation temperature, can be utilized to estimate the nucleation rate and induction time as functions of $T_{ref}^{*}$ and the cooling rate using theoretical expressions. These expressions offer valuable theoretical insights based on the MSZW data.

For this analysis, three commonly used models were applied: the classical Nyvlt expression (see Section 2.3), Sangwal’s self-consistent Nyvlt-like expression (see Section 2.4) and Kubota’s model (see Section 2.5). Additionally, a new theoretical model (see Section 2.5) based on classical nucleation theory was proposed to estimate the nucleation kinetic constant, free energy of nucleation, surface energy and the radius of the critical nucleus. The interpretations and the implications of the MSZW data, calculated using Equations (5) and (6), are thoroughly discussed in the following sections.

### 2.3. Nyvlt’s Approach to the Nucleation Kinetics

According to Nyvlt, the primary nucleation rate is related to the supersaturation by a power law-type expression, given below [32,37]:(7)$$J=k_{n}{\left(∆C_{max}\right)}^{n}$$
where *J* is the nucleation rate (g/L.s), *k*_n_ is the mass nucleation rate proportionality constant (g^1−*n*^/L^1−*n*^.s) and *n* is the apparent order of nucleation. ∆*C*_max_ is the difference between the solubility concentration $C_{T_{ref}^{*}}^{*}\;$at the reference temperature (g/L), $T_{ref}^{*}$, and the solubility concentration $C_{T_{nuc}^{*}}^{*}$ at the temperature where the nucleation is observed, $T_{nuc}^{*}$. Mathematically, Δ*C*_max_ is given by the following expression:(8)$${∆C}_{max}={C_{T_{ref}^{*}}^{*}-C}_{T_{nuc}^{*}}^{*}$$

The nucleation rate is also related to the rate of cooling, *R* = d*T*/d*t*, using the relation below:(9)$$J=R\frac{dC^{*}}{dT^{*}}$$
where $\frac{dC^{*}}{dT^{*}}$ is the change in solubility concentration with respect to temperature, or it can be related to the slope of the solubility curve, *C** vs. *T**.

At the point of nucleation, $\frac{dC^{*}}{dT^{*}}$,(10)$$\frac{dC^{*}}{dT^{*}}=\frac{∆C_{max}}{∆T_{max}}\;$$
where $∆T_{max}$ is the metastable zone width of the API obtained at a given cooling rate. The MSZW is the difference between the reference temperature and the temperature at which nucleation is observed.

Substituting Equation (10) in Equation (7), we obtain(11)$$J=k_{n}{\left(\frac{dC^{*}}{dT^{*}}∆T_{max}\right)}^{n}$$

Substituting Equation (9) for *J* in Equation (11), we obtain(12)$$R\frac{dC^{*}}{dT^{*}}=k_{n}{\left(∆T_{max}\frac{dC^{*}}{dT^{*}}\right)}^{n}$$

Taking log on both sides, we obtain(13)$$\ln\left(R\frac{dC^{*}}{dT^{*}}\right)=\ln\left(k_{n}{\left(∆T_{max}.\frac{dC^{*}}{dT^{*}}\right)}^{n}\right)$$

Equation (13) can be simplified as follows:(14)$$\ln∆T_{max}=\left(\frac{n-1}{n}\right)\ln\left(\frac{dC^{*}}{dT^{*}}\right)+\frac{1}{n}\ln k_{n}+\frac{1}{n}\ln R$$

According to Equation (14), the order of the nucleation, *n*, can be obtained from the slope of $\ln∆T_{max}\;\mathrm{vs}.\;\ln R$; taking the inverse of the slope will yield the value of *n*. Note that if *R* is expressed in terms of the K/s and *dC**/*dT** is expressed in terms of g/L.K, then the *k_n_* obtained from the intercept using Equation (14) will be expressed in terms of (g^1−*n*^/L^1−*n*^.s).

The nucleation rate can be estimated from the intercept based on $\frac{dC^{*}}{dT^{*}}$, as we know the relationship between *C** vs. *T* (see Equation (15)); note that this expression is valid only if the *T^*^* is expressed in terms of K).*C**= 0.007*T**^2^ − 1.8423*T** + 37.563(15)

From Equation (15), we can obtain, d*C^*^*/d*T** as follows:(16)$$\frac{dC^{*}}{dT^{*}}=0.014T^{*}-1.8423$$

Theoretically, it is possible to obtain the nucleation rate as a function of the temperature from the intercept of the plot of $\ln∆T_{max}\;vs.\;\ln R$. The temperature here refers to the reference temperature, $T_{ref}^{*}$.

According to Equation (14), a plot of $\ln∆T_{max}\;vs.\;\ln R$ should allow us to calculate the order of the nucleation from the slope and the nucleation kinetic constant, *k_n_*, from the intercept. In Table 1, we show the determined kinetic constant and the order for the MSZW measured at different $T_{ref}^{*}$. Table 1 also contains the $T_{nuc}^{*}$ obtained using Equations (3) and (4) together with MSZW.

The nucleation rate constant *k_n_* and the nucleation order *n* were determined from the slope and intercept of these plots, with the calculated values presented in Table 1. This table also includes nucleation data for other organic and inorganic compounds, showing that the nucleation order for the studied system is comparable to that of these compounds. As mentioned by Nyvlt et al., while calculating the nucleation parameters, a relatively small change in the value of the slope *n* results in a great change in the value of the kinetic constant *k_n_*. Thus, Nyvlt suggested to consider the values of *k_n_^1/n^* to compensate for such effects. In this work, we found that this value ranges from 4.02 × 10^−3^ to 1.67 × 10^−2^ (g^1−*n*^/L^1−*n*^.s)^1/*n*^. This range of values seems to agree with the ones reported by Nyvlt for several inorganic compounds [37]. As expected, the nucleation rate increases with the cooling rate, while the nucleation order decreases. The nucleation rate also increases with the driving force, irrespective of the cooling rate. These trends can be explained using the classical nucleation theory, which suggests that the nucleation rate is proportional to the supersaturation. Nucleation is chemically controlled and depends on both the driving force and an induction time, which refers to the time required for molecules to form a prenucleation cluster that evolves into a stable nucleus. At higher cooling rates, the solution reaches the critical nucleation temperature more quickly, increasing the driving force for crystallization before the induction time elapses, thus raising the nucleation rate. Typically, the width of metastability in solutions depends on the probability of the solute molecules interacting with each other so that they can form prenucleation clusters or nuclei. The higher the concentration, the higher is the probability of collision and interactions between molecules. In solutions that contain lower concentrations of the crystallising molecules, the collision between the molecules in the solution is relatively low, and the change in formation of prenucleation clusters or nuclei is low. In contrast, at slower cooling rates, the molecules have more time to assemble into stable nuclei at higher temperatures, leading to nucleation occurring at temperatures above those observed under faster cooling. This results in a lower nucleation rate at slower cooling rates, consistent with classical nucleation theory.

### 2.4. Kubota’s Approach

According to Kubota, MSZW is assumed to be a point where the accumulated crystals that are grown up to detectable size have reached a fixed value *N_m_*/*V* at the time *t* = *t_m_* that can be related to the nucleation rate *J* by the relation [34,38]:(17)$$\frac{N_{m}}{V}=\int_{0}^{N_{m}}d\left(\frac{N}{V}\right)=\int_{0}^{t_{m}}Jdt$$

In Equation (17), *t_m_* is the induction time. In Equation (17), the number of nuclei (*N*) accumulated in a unit volume of solution *V* is assumed to be equal to 0 and to *N_m_* when *t* = 0 and *t* = *t_m_*, respectively. According to Equation (17), *J* is expressed in terms of the number of nuclei per unit volume per unit time, as *N_m_* refers to the number of nuclei, *V* is the volume of solvent in L and *t_m_* is expressed in seconds.

Kubota, furthermore, assumed that there exists a linear solubility–temperature relationship. This means that *dC**/*dT* = constant = Δ*C_max_*/Δ*T_max_*. Thus, Kubota defined the nucleation rate *J* as follows [34,38]:(18)$$J=k_{n}{\left(∆T\right)}^{n}$$
where $k_{n}={k_{n^{\prime }}\left(\frac{dC^{*}}{dT^{*}}\right)}^{n}$ [34,38]. In our work, we used Equation (16) to calculate the values of *dC**/*dT** rather than assuming it to be linear and constant, as was assumed by Kubota.

If *J* is expressed in terms of #/L.s and Δ*T* is expressed in terms of K, then the kinetic constant *k_n_* in Equation (18) is expressed in terms of #/L.s.K^n^, *C** is expressed in terms of g/L and *T^*^* is expressed in terms of K, and then, $k_{n}$ will be defined in terms of #.L*^n^*^−1^/s.g*^n^*.

Equation (17) can be rewritten as follows:(19)$$\frac{N_{m}}{V}=\int_{0}^{t_{m}}Jdt=\int_{0}^{t_{m}}Jdt\frac{d(∆T)}{d(∆T)}=\int_{0}^{{∆T}_{max}}J\frac{d\left(∆T\right)}{R}$$
where *R* is the cooling rate, which is defined as *d*(Δ*T*)/*dt*. Substituting Equation (18) in Equation (19) for *J*, we obtain(20)$$\frac{N_{m}}{V}=\int_{0}^{t_{m}}Jdt=\int_{0}^{t_{m}}Jdt\frac{d(∆T)}{d(∆T)}=\int_{0}^{{∆T}_{max}}k_{n}{\left(∆T\right)}^{n}\frac{d(∆T)}{R}$$

Equation (20) can be integrated with respect to the MSZW, assuming $k_{n}$ and *n* are constants.(21)∆Tmax=NmV1knn+11n+1R1/(n+1)

Equation (21) can be linearized as follows:(22)ln∆Tmax=1n+1lnNmV1knn+1+1n+1lnR

According to Equation (22), the order of the nucleation can be obtained from the slope of the plot of ln(Δ*T_max_*) versus ln(*R*), while the nucleation rate constant $k_{n}$ can obtained from the intercept. However, this requires a suitable assumption for the value for the number of nuclei that are grown up to a detectable size per unit volume. This value is typically determined through additional experiments, such as those measuring the induction time. It should be remembered that *N_m_/V* is sensitive to the analytical method used to detect the phase change in the solution and is sensitive to the volume. In this work, we assume a value of 10^8^ #/L, which is a reasonable value for *N_m_/V* and which seems to agree with the experimentally obtained values for this compound. Later, in Section 2.6, we estimate this value theoretically using a mathematical model that we propose.

In Table 2, we show the nucleation rate constant $k_{n}$ and the nucleation order predicted using the Kubota expression as in Equation (22). The order of reaction decreases with the temperature and the kinetic constant increases with the temperature, which agrees with the trends reported in the literature for other systems.

Kubota also defined the induction time, *t_ind_*, which refers to the time in which nucleation can be experimentally detected in a solution subjected to cooling from a saturation temperature at a fixed cooling rate. As nucleation is related to MSZW and induction time, there should exist a relation between MSZW, *t_ind_* and the nucleation rate. Kubota defined induction time as the time required for the number density *N_m_/V* of nuclei that can be experimentally detected to reach a fixed value. The nucleation rate, as in Equation (19), can be related to the number density, *N_m_/V*, as follows [34,38]:(23)$$\frac{N_{m}}{V}=\int_{0}^{t_{ind}}Jdt=\int_{0}^{t_{ind}}Jdt=\int_{0}^{{∆T}_{max}}k_{n}{\left(∆T\right)}^{n}dt$$

At the point of nucleation, Δ*T* is constant (typically, this should be equal to Δ*T_max_*). Equation (23) becomes(24)$$\frac{N_{m}}{V}=k_{n}{\left({∆T}_{max}\right)}^{n}t_{ind}$$

Based on the calculated nucleation rate constant and order, we can mathematically determine the induction time where the phase change is likely to occur using the expression,(25)$$t_{ind}=\frac{N_{m}}{V}\frac{1}{k_{n}}{\left({∆T}_{max}\right)}^{-n}$$

As the constants *N_m_/V*, $k_{n}$ and *n* are the same as in Equation (22), it is possible to obtain additional insight into the induction time, theoretically based on the nucleation rate constant and the order of nucleation. In Figure 1a, we show the *t_ind_* calculated based on the MSZW using Kubota’s expression. The calculated value, *N_m_*/*V*$k_{n}$, for the studied system seems to range from 10^38^ to 10^7^ (K*^n^*.s) or ((°C)*^n^*.s). The calculated *N_m_*/*V*$k_{n}$ value, derived from the MSZW data observed in solutions with lower $T_{ref}^{*}$, aligns well with the values reported in the literature for both organic and inorganic systems [34,38,39,40,41]. The order of nucleation decreases with$\;T_{nuc}^{*}$. The kinetic constant increases with an increase in temperature. The magnitude of the nucleation rate constantly increases from 10^−28^ to 10^3^ #/L.s for an increase in reference temperature from 0 °C to 50 °C. In terms of the induction time obtained theoretically using the Kubota model, irrespective of the cooling rate, the induction time decreases with an increase in the Δ*C_max_* and Δ*T_max_* (see Figure 1a).

Figure 3a shows that the *t_ind_* decreases as the cooling rate increases, which aligns with the theoretical fact that the nucleation rate rises with an increasing driving force (see also Figure 3b). Furthermore, the theoretically obtained *t_ind_* deviates from the linear trend at the lower Δ*T_max_* obtained at the studied cooling rates. In general, there exists globally a linear trend between Δ*T_max_* and *t_ind_*. In terms of the cooling rate, higher cooling rates enhance the nucleation rate by rapidly reaching the critical nucleation point, thereby increasing the driving force when the induction time is reached. Figure 3b clearly shows the relationship between Δ*T_max_* and *J*; the nucleation rate increases linearly with the degree of supercooling, represented by the metastable zone width (Δ*T_max_*). Additionally, *J* is observed to be higher at higher cooling rates. These results can be attributed to the increase in the driving force (Δ*C_max_*) with larger Δ*T_max_*. At higher cooling rates, the solution achieves slightly higher supersaturation at the induction time, which conceptually leads to an increase in the nucleation rate.

### 2.5. Sangwal’s Approach

Sangwal defined the nucleation rate as a function of supersaturation at the point of nucleation by the relation [33](26)$$J=k_{m}{\left(\ln S_{T_{nuc}}\right)}^{m}$$
where $S_{T_{nuc}}$ is the ratio of the solubility concentration of the solute at the reference temperature to the solubility concentration of the solute at the point of nucleation.(27)$$S_{T_{nuc}}=\frac{C_{Tref}^{*}}{C_{T_{nuc}}^{*}}$$

In the above expression, *m* is the order of the nucleation and $S_{T_{nuc}}$ is the dimensionless number, and then, the whole term ${\left(\ln S_{T_{nuc}}\right)}^{m}$ is also a dimensionless number, and *k_m_* has the unit of # nuclei/L.s. Then, the nucleation rate *J* also has the same unit of # nuclei/L.s.

Sangwal used the theory of solutions that correlates the MSZW or $∆T_{max}$, supersaturation ratio $S_{T_{nuc}}$ and the heat of dissolution ${∆H}_{S}$ using the following expression [33]:(28)$$\left(\ln S_{T_{nuc}}\right)=ln\left(\frac{C_{Tref}^{*}}{C_{T_{nuc}}^{*}}\right)=\frac{{∆H}_{S}}{R_{g}T_{nuc}^{*}}\frac{∆T_{max}}{T_{ref}^{*}}$$

We can define the supersaturation ratio using the following relation:(29)$$\sigma_{max}=\frac{∆C_{max}}{C_{T_{nuc}}^{*}}=\frac{C_{Tref}^{*}-C_{T_{nuc}}^{*}}{C_{T_{nuc}}^{*}}=\frac{C_{Tref}^{*}}{C_{T_{nuc}}^{*}}-1=S_{T_{nuc}}-1$$

At the point of nucleation, i.e., at MSZW, one can expect nucleation. At this point, the nucleation rate *J* can be defined as(30)$$J=f\frac{{∆C}_{max}}{C_{T_{nuc}}^{*}∆t}=f\frac{{∆C}_{max}}{C_{T_{nuc}}^{*}∆t}\frac{∆T}{∆T}=f\frac{{∆C}_{max}}{C_{T_{nuc}}^{*}}\frac{R}{∆T}$$
where $\frac{∆T}{∆t}=R$, defined as the cooling rate. In Equation (30), at the point of nucleation, Δ*T* = Δ*T_max_*.

If the assumption $\ln\left(S_{T_{nuc}}\right)=ln\left(\frac{C_{Tref}^{*}}{C_{T_{nuc}}^{*}}\right)\approx \sigma_{max}$ holds true, then Equation (28) becomes(31)$$ln\left(S_{T_{nuc}}\right)=ln\left(\frac{C_{Tref}^{*}}{C_{T_{nuc}}^{*}}\right)=\sigma_{max}=\frac{{∆C}_{max}}{C_{T_{nuc}}^{*}}=\frac{{∆H}_{S}}{R_{g}T_{nuc}}\frac{∆T_{max}}{T_{ref}^{*}}$$

Substituting Equation (29) for $\frac{{∆C}_{max}}{C_{T_{nuc}}^{*}}$ in Equation (30), we obtain(32)$$J=f\frac{{∆H}_{S}}{R_{g}T_{nuc}}\frac{R}{T_{ref}^{*}}$$

From Equations (26) and (32),(33)$$J=f\frac{{∆H}_{S}}{R_{g}T_{nuc}}\frac{R}{T_{ref}^{*}}=k_{m}{\left(\ln S_{T_{nuc}}\right)}^{m}$$

Upon substituting Equation (31) in Equation (33) for $\ln S_{T_{nuc}}$, we obtain(34)$$J=f\frac{{∆H}_{S}}{R_{g}T_{nuc}}\frac{R}{T_{ref}^{*}}=k_{m}{\left(\frac{{∆H}_{S}}{R_{g}T_{nuc}}\frac{∆T_{max}}{T_{ref}^{*}}\right)}^{m}$$

Equation (34) can be rearranged as follows:(35)$$\frac{∆T_{max}}{T_{ref}^{*}}={\left(\frac{f}{k_{m}T_{ref}^{*}}\right)}^{1/m}{\left(\frac{{∆H}_{S}}{R_{g}T_{nuc}}\right)}^{1-m/m}R^{1/m}$$

Equation (35) can be linearized to(36)$$ln\frac{∆T_{max}}{T_{ref}^{*}}=\frac{1}{m}ln\left(\frac{f}{k_{m}T_{ref}^{*}}\right)+\frac{1-m}{m}\ln\left(\frac{{∆H}_{S}}{R_{g}T_{nuc}}\right)+\frac{1}{m}\ln R$$

According to Equation (36), a plot of $ln\frac{∆T_{max}}{T_{ref}^{*}}$ versus ln *R* should theoretically allow us to calculate the order of the nucleation, *m*, from the slope and the nucleation rate constant, *k_m_*, from the intercept. To theoretically estimate the nucleation rate constant, it is essential to calculate the heat of dissolution and assume a suitable value for the value of *f*. Sangwal assumes *f* is related to the concentration of the prenucleation clusters, and Sangwal recommended using the value of *C*N_A_*, where *N_A_* is Avogadro’s number and *C^*^* is the solubility concentration expressed in terms of moles/litre. For the studied system and for the range of $T_{ref}^{*}$ considered in this work, we found that the *f* values were approximately equal to 10^23^ molecules/L. In addition to the approximately estimated values for *f*, we theoretically calculated the heat of solution from the solubility data using van’t Hoff’s expression. To obtain the kinetic constant, we used the Δ*H_s_* = 16.28 kJ/mol obtained from the solubility data obtained using the in situ FTIR (see Section 2.1). In Table 1, we show the order of nucleation and the nucleation rate constant obtained from the slope and intercept of ln (Δ*T_max_*/$T_{ref}^{*}$) versus *ln R* using Equation (25), respectively. From Table 1, it is evident that both the nucleation kinetic constant and the order of the nucleation as in the Sangwal expression seems to decrease with an increase in the starting temperature. The kinetic constant ranges from 10^22^ to 10^21^ nuclei/L.s, and the order of the nucleation seems to range approximately from 11 to 2. A similar trend between the kinetic constant and order of nucleation with the temperature was observed for other organic/inorganic compounds [34,38,41,42,43,44].

### 2.6. Our Kinetic Model: Homogeneous 3D Nucleation Model

In Section 2.2 to Section 2.3, we studied the MSZW data using three different theoretical models that assume that the kinetic constant and the nucleation rate are independent of the cooling rate. However, it should be recognized that nucleation is sensitive to the process conditions, especially the particular cooling rate, as it decides ultimately the induction time and the supersaturation concentration where the phase change occurs. Thus, to estimate the nucleation rates at varying cooling rates, we introduce a mathematical model that relies directly on the classical nucleation theory. This model enables the prediction of nucleation rates and Gibbs free energy of nucleation from the metastable zone width (MSZW) measured against solubility temperatures. Unlike previous models, it calculates nucleation rates across different cooling rates.

According to the classical nucleation theory, the nucleation rate *J* is defined as(37)J=kv.exp⁡−ΔGRT

At the point of nucleation, *T* = $T_{nuc}^{*}$,(38)J=kv.exp⁡−ΔGRTnuc*

Expressing *J* in terms of the cooling rate, $R^{\prime }=\frac{dT}{dt}$ and the slope of solubility curve $\frac{dC^{*}}{dT}$,(39)$$J=R^{\prime }\cdot \frac{dC^{*}}{dT}$$

From Equations (30) and (39), it is possible to write that(40)kv.exp⁡−ΔGRTnuc*=R′⋅dC*dT

At the point of nucleation,(41)$$\frac{dC^{*}}{dT}=\frac{∆c_{max}}{∆T_{max}}$$
where $\Delta T_{max}=T_{ref}^{*}-T_{nuc}^{*}$ is the MSZW and ∆*C_max_* is the supersaturation (or the driving force required for crystallization) achieved at $T_{nuc}$, where primary nucleation is most likely to occur.

Substituting Equation (41) in Equation (40), we obtain(42)kv.exp⁡−ΔGRTnuc*=R′∆Cmax∆Tmax

Equation (42) can be linearized as follows:(43)$$\ln\left(\frac{∆C_{max}}{∆T_{max}}\right)+ln\;R^{\prime }=\ln\left(k_{v}\right)-\frac{\Delta G}{RT_{nuc}^{*}}$$

A plot of ln(Δ*C_max_*/Δ*T_max_*) $+\;ln\;R^{\prime }$ versus 1/*T_nuc_* should yield a straight line with a negative slope whose value is equal to Δ*G*/*R* with an intercept equal to ln(*k_n_*). Once the nucleation kinetic constant, *k_v_*, and the Δ*G* are determined, then, mathematically, it is possible to estimate the nucleation rate *J* using Equation (38) at a fixed cooling rate *R’* and the MSZW measured at different $T_{ref}^{*}$ (and the corresponding $T_{nuc}^{*}$). Then, the predicted kinetic constant *k_v_* will be expressed in terms of molecules/m^3^.s, and Δ*G* will be expressed in terms of J/mol.

From the classical nucleation theory using the the surface energy, or the interfacial tension associated with the formation of a stable nucleus Δ*G* is given by [33](44)$$\frac{∆G}{RT_{nuc}^{*}}=\frac{16\pi \gamma^{3}\vartheta^{2}}{3R^{2}{T_{nuc}^{*}}^{2}{\left(\ln S\right)}^{2}}\frac{1}{RT_{nuc}^{*}}$$

Once the surface energy is calculated using Equation (44), we can mathematically calculate the radius of the critical nucleus using the following expression [33]:(45)$$r_{c}=\frac{2\gamma \vartheta }{kT_{nuc}^{*}lnS}$$

For convenience, while implementing our model as in Equation (45), we expressed Δ*C_max_* in terms of molecules/m^3^, Δ*T_max_* in K, *T* in K and the cooling rate *R* in K/s. *T* in Equation (45) typically refers to the nucleation temperature, $T_{nuc}^{*}$. It is worth mentioning here that the new mathematical expression proposed in Equation (43) is obtained from the classical nucleation theory. This means the model does not deviate from the classical nucleation mechanism, which assumes that the solute molecules aggregate into clusters from which three-dimensional stable nucleus evolve into a bulk crystal via homogeneous nucleation in contrast with the two-step nucleation theory. The mathematical expression, as in the Equation (43) model, provides a comprehensive framework to predict the nucleation behaviour at different cooling rates.

In Figure 4, we show the plot of $\ln\left(\frac{∆c_{max}}{∆T_{max}}\right)+ln\;R^{\prime }$ versus $\frac{1}{T_{nuc}^{*}}$ during the crystallization of paracetamol in isopropanol at two different cooling rates. Using Figure 4, the kinetic constant (*k_v_*) and the Gibbs free energy of nucleation (Δ*G*) were calculated from the intercept and slope, respectively, using Equation (43). The plot clearly shows a linear trend and an excellent fit with the experimentally obtained data (as evidenced by an *R^2^* value of 0.99). This points to the fact that the proposed model well represents the experimental data at the studied cooling rates. The calculated *k_v_* and the Δ*G* are given in Table 2. From the Δ*G*, we calculated the nucleation rate, surface free energy and the radius of the critical nucleus using Equation (38), (44) and (45), respectively. The calculated *γ* and *r_c_* as a function of $T_{ref}^{*}$ and at the studied cooling rates, together with the corresponding driving force (expressed in terms of *S*), are given in Table 2. The nucleation constant was found to be in the range of 10^21^ to 10^22^ molecules/m^3^.min, depending on the cooling rate. The free energy of nucleation for paracetamol was found to be equal to 3.62 and 3.63 kJ/mol for the cooling rates 0.01 K and 0.05 K/min, respectively. These values agree with the ones reported (but obtained using the classical nucleation theory) for organic compounds [12,15,45]. The *J*, which is proportional to the Δ*G*, was found to be equal to ~10^22^ molecules/m^3^.s and 10^23^ molecules/m^3^.s. These values agree with the *J* values reported for the organic compounds but predicted using the classical nucleation theory [15,18,45,46]. The surface energy for paracetamol in isopropanol ranges from 2 to 8 mJ/m^2^ and generally decreases with increasing temperature. This trend is expected due to the reduced viscosity and enhanced molecular collisions at the solid–liquid interface at higher temperatures. Similar trends and magnitudes have been observed for other systems, particularly organic compounds [15,18,42,43,47]. The critical radius of the stable nucleus was approximately 10^−10^ m, with the stable nucleus containing about 1 to 3 unit cells.

The *r_c_* values obtained using our model align well with the magnitudes reported for other APIs calculated through different theoretical models [42,48]. The theoretically predicted induction time generally decreases with increasing $T_{ref}^{*}$. For a change in $T_{ref}^{*}$ from 10 °C to 50 °C, the induction time varies significantly by an order of magnitude. Notably, the theoretical *t_ind_* derived from our model matches exactly with the experimentally measured induction times across all conducted experiments, regardless of the set cooling rate.

## 3. Materials and Methods

Acetaminophen (paracetamol) of 99.9% purity was purchased from Sigma-Aldrich and used without further purification. The solvent HPLC grade isopropanol was purchased from Honeywell and used without any further purification.

### 3.1. Solubility Experiment Assisted by Process Analytical Technology Tool, In Situ FTIR

Solubility experiments were performed using an Easymax 102 workstation, and the process was fully monitored and controlled using the Mettler Toledo’s iControl software v4.3. Three different recipes (Recipe 1 to Recipe 3) were created and executed using the iControl software v4.3. For all three recipes, we used a 100 mL reactor with a reactor volume of 70 mL for the solubility measurements. For Recipe 1, exactly 55.02 g of isopropanol was added to the reactor, and the reactor temperature was set to −2 °C. Once the reactor temperature reached −2 °C, we added 13 g of paracetamol (accurately weighed) into the reactor, and then the reactor temperature was maintained at −3 °C for 10 min. After 10 min, we started to increase the reactor temperature to 65 °C at a very slow heating rate of 0.05 °C/min. Once the solution temperature reached the final temperature of 65 °C, the solution was then maintained at this temperature for 40 min (also called the ‘*wait time*’). After the ‘*wait time*’, the solution was then cooled back to a temperature of 30 °C at a cooling rate of 0.05 K. The last step was performed, essentially, to recrystallize the paracetamol and to recover the maximum amount of the dissolved paracetamol as solid.

For Recipe 2, exactly 55.02 g of isopropanol (equivalent to 70 mL) was added to the 100 mL reactor. Then, the reactor was set to −8 °C. Once the solvent reached −8 °C, we added 13 g of paracetamol into the reactor, and then the solution temperature was maintained at this temperature for 10 min. Then, we heated the solution to 65 °C at a slow heating rate of 0.05 K/min. Once the solution reached 65 °C, we maintained the solution at this temperature for 45 min. Then, we cooled the solution back to 30 °C to recover some of the paracetamol dissolved in solution via recrystallization. As mentioned above, the last step is optional.

For Recipe 3, we added 55.02 g of isopropanol to the reactor. Then, we decreased the solution temperature to −2 °C. Once the reactor temperature reached −2 °C, we added 12.7696 g of paracetamol, and then we maintained the reactor at this temperature for 10 min. Then, we rapidly heated the solution to 30 °C as fast as possible. Once the solution reached 30 °C, we maintained the reactor at this temperature for 10 min. Then, we heated the solution to 35 °C at a heating rate of 0.05 K/min. Once the solution reached 35 °C, we maintained the reactor temperature for 120 min. Then, we heated the solution to 50 °C as fast as possible. Once the solution reached 50 °C, we maintained the reactor at this temperature for 10 min. Then, we started to heat the solution to 55 °C at a heating rate of 0.05 K. Once the solution reached this temperature, we maintained the solution at this temperature for 120 min.

The SOPs or recipes described above were created and executed using Mettler Toledo’s iControl software. Throughout the experiment, the solution concentration was monitored using an in situ ATR–FTIR (model: Mettler Toledo’s ReactIR 10 connected to a DiComp probe via AgX 9.5mm × 1.5m Fiber (Silver Halide)). Each spectrum was collected from an average of 64 scans. iC IR (v4.3) software (Mettler Toledo) was used to control the ATR–FTIR and to monitor and collect the spectra with a resolution of 8 cm^−1^ in the wavenumber ranging from 650 to 3000 cm^−1^. In Figure 5a, we show the FTIR spectra of the paracetamol solution at wavenumbers ranging from 650 to 1650 cm^−1^. In Figure 5a, we also show the IR spectra of the pure solvent isopropanol. The FTIR spectra show characteristic peaks for paracetamol at 1516 cm^−1^. The iC IR software was used to measure the height of the characteristic peak that corresponds to the paracetamol; it was obtained by measuring the height of the peak at 1516 cm^−1^ with respect to a two-point baseline that connects 1510 cm^−1^ and 1540 cm^−1^ in the FTIR spectra. During the experiment, the height of the peak at 1516 cm^−1^ monitored using iC IR software was shared with the iControl software, so that all the process variables, such as agitation speed, reaction temperature and the data obtained from PAT, appear in a single analysis window. At the end of the experiment, the observed peak intensities at different reactor temperatures were extracted from the iControl software and then used to obtain the solubility using the procedures described below.

Using the height of the characteristic peak of paracetamol, we developed a one-point calibration method to correlate the FTIR peak height with the solubility concentration [4]. The one-point calibration method relies on the following simple expression to obtain the solubility concentration of paracetamol as a function of the peak height.(46)$$C^{*}=\frac{I_{f}-I_{t}}{I_{f}-I_{o}}\times \left(\frac{M_{o}}{V}-C_{T_{k}}^{*}\right)-\left(C_{f}-C_{T_{k}}^{*}\right)$$

In Equation (46), *M*_o_ refers to the mass of paracetamol added initially to the reactor, and *V* is the volume of the solvent. The ratio of *M_o_*/*V* is the concentration of the solution when all the mass of paracetamol is dissolved in solvent at temperature *T_f_*. The term $C_{T_{k}}^{*}$ refers to the solubility of paracetamol at some known temperature $T_{k}$. This known temperature, $T_{k}$, is usually the temperature at which the intensity of the solution is equal to *I_o_*. To predict solubility using Equation (46), it is essential to know the solubility of paracetamol at least at two known temperatures. The relationship between the parameters involved in Equation (46) is shown in Figure 5b. In our case, we used a value of $C_{T_{k}}^{*}$ = 55.869 g/L, which is equal to the solubility of paracetamol at 0 °C (taken from the work of Granberg [35]), while we converted the intensity to the solubility concentration for Recipe 1. The relationship between the parameters involved in Equation (46) is given in Figure 5b. For the case of Recipe 2, to use Equation (46) we used a value of $C_{T_{k}}^{*}$ = 41.46 g/L, which is equal to the solubility of paracetamol at −7.98 °C (note that this value was obtained from our own PAT-obtained values using Recipe 1). It should be noted that Equation (46) can be used to predict the solubility only if the *C_o_* mass of paracetamol is completely dissolved at some specific temperature during the course of the solubility experiment. The temperature at which all the solids in the reactor of mass *M_o_* are dissolved can be taken as the final temperature *T_f_*. The intensity of the IR peak observed at this temperature can be taken as *I_f_*. The term *I_t_* in Equation (46) refers to the intensity of the FTIR peak observed at any temperature. *I_o_* is the IR peak intensity when time *t* = 0. In our case, *I_o_* always refers to the IR peak intensity measured at temperature $T_{k}\;$, where the solubility (i.e., $C_{T_{k}}^{*}$) is known a priori. Specifically, for Recipe 1 and Recipe 2, *I_o_* corresponds to the IR intensity when the solution temperature is equal to 0 °C and −7.98 °C, respectively. Furthermore, the solubility concentration can be determined using Equation (46) provided we know the temperature, *T_f_*, at which all the mass in the reactor will be dissolved and the corresponding intensity of the FTIR peak observed at 1516 cm^−1^. In other words, mathematically, to use Equation (46), the solubility of paracetamol at at least at two different temperatures must be known a priori (temperature when *I* = *I_o_* and *T_f_*). Mathematically, Equation (46) is valid irrespective of the starting temperature, as irrespective of the starting temperature, *I_t_* is always greater than *I_o_*. When *T* = *T_f_*, all the paracetamol crystals added to the solution will be completely dissolved. The mass of paracetamol dissolved at this point should be equal to *C_o_* (i.e., the mass of the crystals initially added to the solvent in the reactor). In other words, when the solubility temperature *T** = *T_f_*, the solubility concentration will be equal to C_f_.

According to Equation (46), when the solubility temperature *T** = *T*_f_, *I*_t_ = *I*_f_; thus, *C** = *C*_f_ (as (*I*_f −_
*I*_t_)/(*I*_f −_
*I*_o_) = 0). Likewise, when *T** = $T_{k}$, the FTIR intensity recorded at this temperature, *I*_t_ = *I*_o_, and thus, *C** = $C_{T_{k}}^{*}$ (as (*I*_f −_
*I*_o_)/(*I*_f −_
*I*_o_) = 1). Clearly, Equation (1) predicts the solubility concentration $C_{T_{k}}^{*}$ at the temperature $T_{k}$ and when *T* = *T*_f_. It should be noted that the solubility concentration at *T*_f_ is experimentally obtained based on the FTIR intensity, whereas $C_{T_{k}}^{*}$ can be obtained using the gravimetric method, and in the present case, this is taken from the literature. As Equation (46) accurately predicts the solubility values at these two limits, we propose that this expression should accurately predict the solubility of the paracetamol at temperatures that range between $T_{k}$ and *T*_f_.

### 3.2. Metastable Zone Width Prediction Using PAT

The MSZW was obtained using a polythermal method [36]. To obtain the MSZW at a reference solubility temperature, $T_{ref}^{*}$, we dissolved a known mass of paracetamol which is equivalent to the solubility concentration of paracetamol, *C**, in isopropanol at $T_{ref}^{*}$. Then, we heated the solution to 65 °C at a much faster rate, and then we maintained the solution at this temperature for 45 min. This step was purposely included to ensure complete dissolution of the solute in the solvent. Then, we cooled the solution to $T_{ref}^{*}$ + 5 °C at a much faster rate. This step was introduced to maintain the experimental consistency, so that all the MSZW experiments began from a temperature that is equal to $T_{ref}^{*}$ + 5 °C. Then, we cooled the solution from $T_{ref}^{*}$ + 5 °C to $T_{ref}^{*}$ at a fixed cooling rate of 0.05 K/min. Once the solution reached the reference solubility temperature, $T_{ref}^{*}$, we cooled the solution at a fixed cooling rate of 0.01 K/min or 0.05 k/min until we observed the nucleation. In this work, we measured the MSZW at four different $T_{ref}^{*}$ values, 14.8 °C, 23.22 °C, 33.4 °C and 44.47 °C, at a cooling rate of 0.01 K/min and at three different $T_{ref}^{*}$ values, 23.67 °C, 33.65 °C, and 43.63 °C, at a cooling rate of 0.05 K/min. We define the MSZW at *T*^*^ as the difference between the *T*^*^ and the temperature at which we observed the nucleation, $T_{nuc}^{*}$. Mathematically, MSZW can be written as [36]:(47)$$\mathrm{MSZW}={∆T_{m}=T}_{ref}^{*}-T_{nuc}^{*}$$

The point of nucleation was monitored using the PAT tool, in situ FBRM. We were monitoring the total counts. As the measurement of the MSZW relied on the FBRM, and considering the limitations of the instrument, we assumed that, once the stable nuclei were formed, it instantaneously triggered the secondary nucleation, thus increasing the counts that could be detected using FBRM. Furthermore, we assumed the time at which the first nuclei should have formed and the time at which we could detect the nucleation using FBRM is too small, and thus was negligible. Once we determined the MSZW, we calculated the supersolubility concentration at different $T_{nuc}^{*}$ with respect to the $T_{ref}^{*}$ (see Section 2.1 for details).

## 4. Conclusions

To conclude, we reported the solubility and metastable zone width (MSZW) data for paracetamol in isopropanol, obtained using advanced process analytical technology (PAT) tools, specifically, in situ Fourier Transform Infrared (FTIR) spectroscopy and Focused Beam Reflectance Measurement (FBRM). We also presented the standard operating procedures (SOPs) for data acquisition and processing, detailing methods to convert spectroscopic data and FBRM counts into the solubility concentration and the metastable zone width or the supersolubility concentration. The MSZW broadens with increased cooling rates regardless of the solubility temperature. This observation aligns with experimental evidence showing that higher cooling rates lead to elevated supersolubility concentrations at corresponding nucleation temperatures compared with those observed at lower cooling rates. The MSZW data were analysed using the three different theoretical models, in addition to a new model proposed by us, to obtain the kinetic and thermodynamic parameters that are relevant to the nucleation. The proposed model excellently represents the experimental data and, more importantly, allows us to predict the nucleation kinetic constant and the Gibbs free energy at different cooling rates.

The proposed model excellently fits the experimental data and successfully predicts the nucleation rate, Gibbs free energy of nucleation, surface energy and the size of the critical nuclei at different cooling rates as a function of solubility temperature or the maximum supersaturation at the nucleation temperature.

## Figures and Tables

**Figure 1 pharmaceuticals-18-00314-f001:**
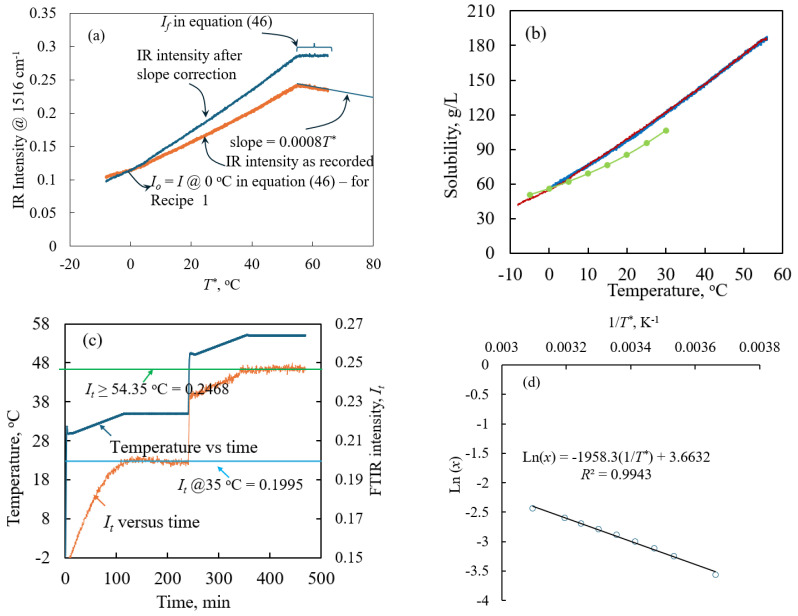
(**a**) IR intensity versus the solubility temperature (– IR intensity as recorded by the PAT using Recipe 1, –: IR intensity after correcting the temperature effects on the recorded IR intensity). (**b**) Solubility of paracetamol measured at different temperatures (–: PAT-obtained solubility using Recipe 1, – PAT-obtained solubility using Recipe 2, ●: solubility taken from the literature [35]). (**c**) Plot of temperature (–) and FTIR intensity (–) versus time. (**d**) Plot of ln (*x*) versus 1/*T**.

**Figure 2 pharmaceuticals-18-00314-f002:**
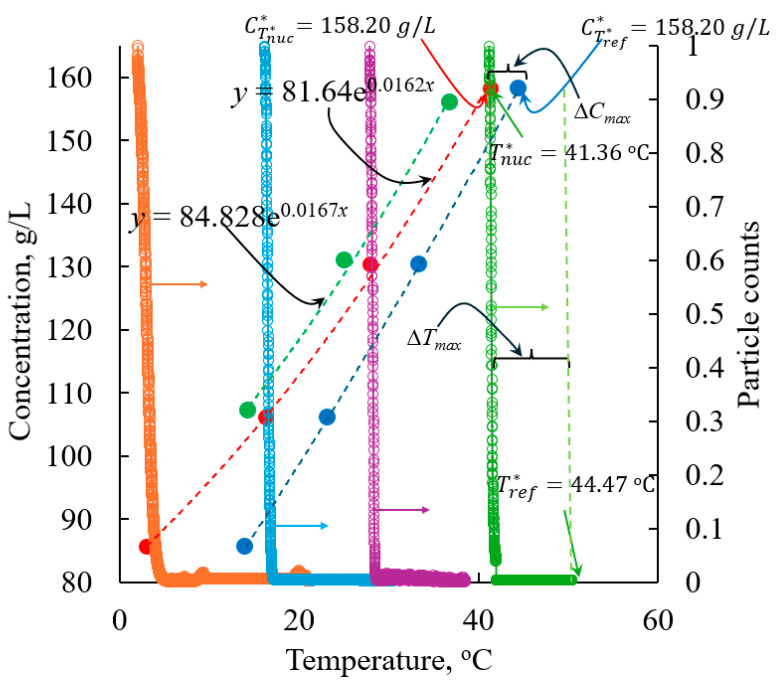
Concentration measured using the FTIR and the normalized particle counts measured using the FBRM versus temperature. ●: Solubility obtained using FTIR at the studied $T_{ref}^{*}$; ●: MSZW obtained at a cooling rate of 0.01 K/min; ●: MSZW obtained at a cooling rate of 0.05 K/min; ○: normalized particle count measured during the MSZW experiment for $T_{ref}^{*}$ = 44.47 °C; ○: normalized particle count measured during the MSZW experiment for $T_{ref}^{*}$ = 33.4 °C; ○: normalized particle count measured during the MSZW experiment for $T_{ref}^{*}$ = 23.22 °C; ○: normalized particle count measured during the MSZW experiment for $T_{ref}^{*}$ = 14.8 °C; ▬ ▬
*C^*^* versus *T^*^*; ▬ ▬
*c_s_* versus $T_{ref}^{*}$ at d*T*/d*t* = 0.01 K/min; ▬ ▬
*c_s_* versus $T_{ref}^{*}$ at d*T*/d*t* = 0.05 K/min. Note: For the convenience of the readers, we also show the relationships between $T_{ref}^{*}$, $T_{nuc}^{*}$, ∆*T_max_* and *c_s_*.

**Figure 3 pharmaceuticals-18-00314-f003:**
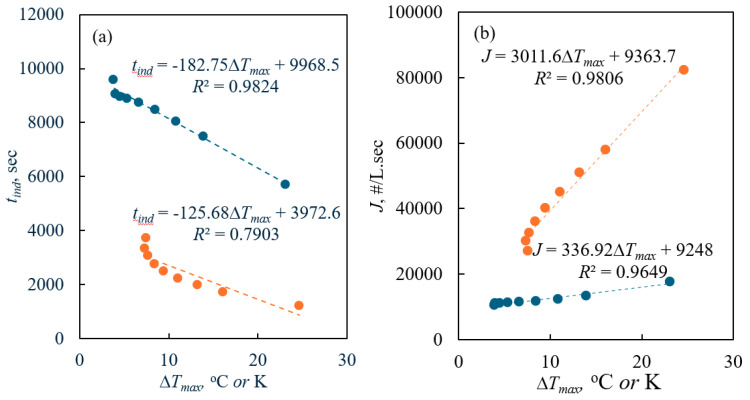
(**a**) Plot of the induction time obtained theoretically using Kubota’s model versus the experimentally measured Δ*T_max_
*(●: *R’*: 0.01 K/min, ●: *R’*: 0.05 K/min; Dotted lines are provided as a guide to the eye); (**b**) plot of the nucleation rate calculated theoretically using Kubota’s model versus the Δ*T_max_* (●: *R’*: 0.01 K/min, ●: *R’*: 0.05 K/min; Dotted lines are provided as a guide to the eye).

**Figure 4 pharmaceuticals-18-00314-f004:**
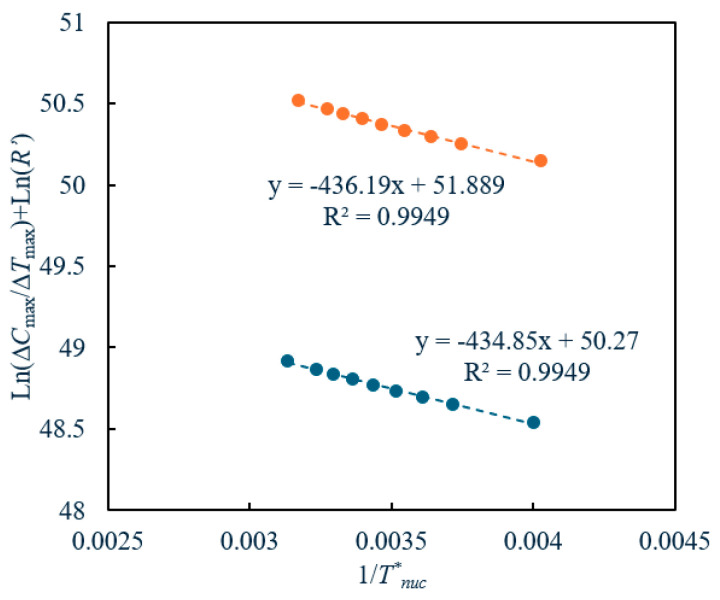
A plot of ln(Δ*C_max_*/Δ*T_max_*) + $ln\;R^{\prime }$ versus 1/*T_nuc_* according to our model (●: *R’*: 0.01 K/min, ●: *R’*: 0.05 K/min).

**Figure 5 pharmaceuticals-18-00314-f005:**
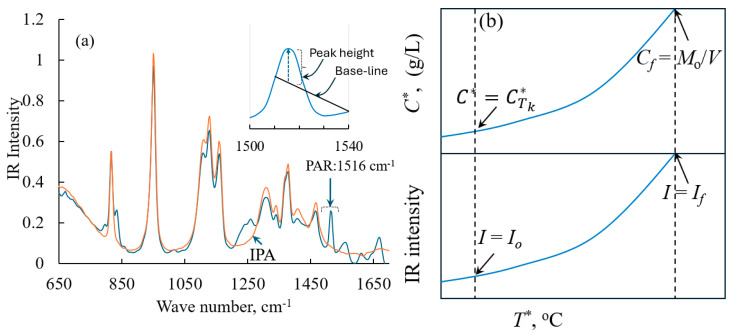
(**a**) FTIR spectra of the paracetamol solution (blue line) and pure isopropanol (orange line). The figure insert illustrates the method used to calculate the peak height during the crystallization process. (**b**) Plot of temperature versus the IR intensity and the solubility concentration. In the figure, we marked different parameters to show the relationship between the parameters involved in Equation (46) and solubility concentration *C**.

**Table 1 pharmaceuticals-18-00314-t001:** Solubility concentration at $T_{ref}^{*}$ and $T_{nuc}^{*}$, the MSZW measured at different cooling rates, the calculated nucleation parameters and the nucleation rate based on the Nyvlt expression.

Initial Conditions		Cooling Rate: 0.01 K/min			Cooling Rate: 0.05 K/min			Nucleation Parameters (Nyvlt)				Nucleation Parameters (Kubota)		Nucleation Parameters (Sangwal)	
$T_{ref}^{*}$	*C* ^*^	$T_{nuc}^{*}$	$C^{*}@$ $T_{nuc}^{*}$	Δ*C_max_*	$T_{nuc}^{*}$	$C^{*}@$ $T_{nuc}^{*}$	Δ*C_max_*	*n*	*k_n_*	*k_n_^1/n^*	*J*	*n*	*k_n_*	*k_m_*	*n*
(°C)	(g/L)	(°C)	(g/L)	(g/L)	(°C)	(g/L)	(g/L)	-	(g^1−n^/L^1−n^.s)	(g^1−n^/L^1−n^.s)^1/n^	(#/m^3^.min)	-	(#/L.K^n^.s)	(#/L.s)	-
0	56.32	−23.09	14.33	41.99	−24.67	11.74	44.58	24.39	1.06 × 10^−44^	1.57 × 10^−2^	4.25 × 10^22^	23.39	2.25 × 10^−28^	2.42 × 10^22^	24.39
10	76.82	−3.94	48.64	28.18	−6.08	44.53	32.28	11.22	1.11 × 10^−20^	1.67 × 10^−2^	2.28 × 10^23^	10.22	2.69 × 10^−8^	1.31 × 10^23^	11.22
15	87.59	4.17	64.69	22.90	1.77	59.85	27.74	8.06	2.97 × 10^−15^	1.58 × 10^−2^	3.08 × 10^23^	7.06	6.09 × 10^−4^	6.25 × 10^22^	8.06
20	98.71	11.55	80.11	18.60	8.93	74.56	24.15	5.97	8.39 × 10^−12^	1.40 × 10^−2^	3.66 × 10^23^	4.97	2.89 × 10^−1^	2.41 × 10^22^	5.97
25	110.19	18.33	94.97	15.22	15.52	88.72	21.46	4.57	1.41 × 10^−9^	1.15 × 10^−2^	4.07 × 10^23^	3.57	1.32 × 10^1^	9.45 × 10^21^	4.57
30	122.01	24.63	109.32	12.69	21.62	102.39	19.61	3.62	3.79 × 10^−8^	8.95 × 10^−3^	4.37 × 10^23^	2.62	1.37 × 10^2^	4.23 × 10^21^	3.62
35	134.18	30.50	123.20	10.98	27.32	115.62	18.56	3.01	2.91 × 10^−7^	6.76 × 10^−3^	4.6 × 10^23^	2.01	5.40 × 10^2^	2.29 × 10^21^	3.01
40	146.71	36.01	136.67	10.03	32.66	128.44	18.27	2.65	9.21 × 10^−7^	5.23 × 10^−3^	4.79 × 10^23^	1.65	1.13 × 10^3^	1.53 × 10^21^	2.65

**Table 2 pharmaceuticals-18-00314-t002:** Kinetic and thermodynamic parameters for paracetamol crystallization in isopropanol, predicted using our homogeneous 3D model based on the MSZW.

Kinetics, thermodynamics and parameters relevant to nucleation for MSZW obtained at a cooling rate of 0.01 K/min									
*T*, °C	*k_v_*, #/m^3^.s	*S* = *C*/*C^*^*	ΔG, kJ	J, #/m^3^.s	*γ*, mJ/m^2^	*r_c_*, m	# of u.c. in *r_c_*	*N_m_/V*,#/m^3^	*t_ind_*, s
0	6.79 × 10^21^	4.80	3.62	3.87 × 10^22^	14.81	6.28 × 10^−10^	1	5.88 × 10^27^	29,605
10		1.72		3.42 × 10^22^	8.02	9.45 × 10^−10^	2	3.39 × 10^27^	19,302
15		1.46		3.26 × 10^22^	6.38	1.10 × 10^−9^	2	2.66 × 10^27^	15,871
20		1.32		3.13 × 10^22^	5.18	1.26 × 10^−9^	2	2.14 × 10^27^	13,280
25		1.24		3.02 × 10^22^	4.29	1.43 × 10^−9^	3	1.77 × 10^27^	11,378
30		1.19		2.93 × 10^22^	3.62	1.61 × 10^−9^	3	1.52 × 10^27^	10,054
35		1.16		2.85 × 10^22^	3.15	1.76 × 10^−9^	3	1.35 × 10^27^	9220
40		1.14		2.77 × 10^22^	2.85	1.88 × 10^−9^	3	1.26 × 10^27^	8808
50		1.13		2.65 × 10^22^	2.67	1.97 × 10^−9^	3	1.24 × 10^27^	9044
Kinetics, thermodynamics and parameters relevant to nucleation for MSZW obtained at a cooling rate of 0.05 K/min									
*T*, °C	*k_v_*, #/m^3^.s	*S*		*J*, #/m^3^.s	*γ*, mJ/m^2^	*r_c_*, m	# of u.c. in *r_c_*	*N_m_/V*,#/m^3^	*t_ind_*, s
0	3.43 × 10^22^	4.80	3.63	1.99 × 10^23^	16.03	5.97 × 10^−10^	1	5.88 × 10^27^	29,605
10		1.72		1.76 × 10^23^	8.88	8.84 × 10^−10^	2	3.39 × 10^27^	19,302
15		1.46		1.68 × 10^23^	7.28	1.01 × 10^−9^	2	2.66 × 10^27^	15,871
20		1.32		1.61 × 10^23^	6.16	1.13 × 10^−9^	2	2.14 × 10^27^	13,280
25		1.24		1.55 × 10^23^	5.35	1.24 × 10^−9^	2	1.77 × 10^27^	11,378
30		1.19		1.51 × 10^23^	4.77	1.34 × 10^−9^	2	1.52 × 10^27^	10,054
35		1.16		1.47 × 10^23^	4.37	1.42 × 10^−9^	3	1.35 × 10^27^	9220
40		1.14		1.43 × 10^23^	4.13	1.47 × 10^−9^	3	1.26 × 10^27^	8808
50		1.13		1.37 × 10^23^	3.99	1.51 × 10^−9^	3	1.24 × 10^27^	9044

## Data Availability

The solubility and MSZW data are available from the authors upon request. Additionally, the recipe that can be executed using iControl software can be provided upon request to measure the solubility and MSZW using the Mettler Toledo workstation and PAT that can be integrated into the workstation.

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
