# Peer review of "Process Analytical Technology Obtained Metastable Zone Width, Nucleation Rate and Solubility of Paracetamol in Isopropanol—Theoretical Analysis"

_pharmaceuticals, 2025, doi:10.3390/ph18030314_

Round 1
Reviewer 1 Report
Comments and Suggestions for Authors
Article “Process analytical technology obtained metastable zone width, nucleation rate and solubility of paracetamol in isopropanol – theoretical analysis” by Mahmoud Ranjbar, Mayank Vashishtha, Gavin Walker, and K Vasanth Kumar describes use of process analytical technology to obtain solubility information, MSZW and nucleation parameters. Authors initially designed experimental approach to obtain reliable information about solubility, then used the obtained data to design experiments for measuring MSZW at different temperatures and then used several approaches to obtain nucleation kinetic parameters, including an approach developed by the authors. Overall, the article is well written, although the description is very detailed and, in several places, seems to be too detailed and providing information which could be easily excluded from the article. Therefore, the text could be shortened by making the article more appealing for the readers.
Nevertheless, some comments and objections regarding the article appeared.
1. Line 245 (L245) Authors write that exponential relationship exists, but equation (3) provides polynomial relationship.
2. The authors use terms “solubility concentration” and “solubility temperature”. Although the meaning can be understood, could some better term be used instead?
3. L313 The unit kJ/(mol·K) seems much too large for entropy, magnitude of which rarely exceeds hundreds of J/(mol·K). Check whether the number and the unit are correct.
4. At the beginning of Section 3.2. (L315) Figure 3 is described. The Figure 3 appearing several pages below presents different data. It seems that the mentioned Figure 3 is missing from the article.
5. Supersaturation is often defined as the ratio between the concentration in the solution and the solubility at the particular temperature (as used in equation (29)). In the text supersaturation is also used to describe the difference of these concentrations ΔC (as in equation (10)). This should be clarified.
6. The term supersolubility concentration used by the authors should be clarified. I have not seen such term previously. How does it different from e.g. supersaturation?
7. MSZW is determined using only one measurement under each experimental condition. As the nucleation is a kinetic event, the nucleation of a solution with a given concentration is likely to spread over a temperature range. Is it a common practice that no repeated measurements are performed for such measurements?
8. On L428 authors write that a Kubota model assumes a linear solubility-temperature relationship. This seems wrong, as instead exponential relationship exists, as also observed in the experiments.
9. In Figure 3 (page 13) nonlinear relationships are presented for cooling rate 0.05 K/min. Nevertheless, linear trendline and equation is used for processing these data and no comments about the deviation from linear relationship mentioned and no explanation provided. Besides, the statistical data are not formatted accordingly (too many significant figures for correlation coefficient and constants of the regression equation.
There are also few minor comments regarding the article text:
10. L42-43 The comment in the parentheses seems to be meant for deletion in the final version.
11. L138. cm-1 left unformatted.
12. In caption of Figure 2 the (a) is duplicated and captions for figure parts (b), (c) and (d) are not appropriate.
13. Relationship in Figure 2(d) show some minor nonlinear characteristic, as first and last points are below the linear trendline, whereas the middle ones are above. Even if not particularly important, maybe this can be mentioned.
Reviewer 2 Report
Comments and Suggestions for Authors
The authors provide a new formula to study the nucleation dynamics. However, I do not think it is a new model. J express in the formula is similar to other approaches, and it directly gives the exponential problem in other approaches, and some optimization suggestions can be given from this aspect.
The structure of the article has serious problems, which affects the readers' reading. It is recommended to adjust the structure of the article before resubmition.
Here are some specific questions:
Q1. The novel model developed based on classical nucleation theory is novel model? Or just Optimized Model? It is different.
Equation 47-48 lacked.
Caption in Figure 2 is not right.
There are also many formatting issues.
Q2. Figure 1a, the baseline which connects 1510 cm-1 and 1540 cm-1 in the FTIR spectra is confused. The height of the characteristic peak of paracetamol can be schemed in Figure 1a.
Q3. In situ IR spectra including bands and intensities are highly influenced by the temperature. How to induce the influences? It is a challenge in our experiments to use IR in Quantitative Analysis.
Q4. The fitting results in the three models were only given in Figure 3 Kubota’s model. Where are the other results?
Comments on the Quality of English LanguageThe English could be improved to more clearly express the research.
Reviewer 3 Report
Comments and Suggestions for Authors
1-Please flowchart or step-by-step outline of the experimental setup and data analysis process would enhance clarity and allow readers to better understand the procedures involved in measuring MSZW and solubility. This could also facilitate reproducibility of the experiments by other researchers.
2-The moisture content in solvent isopropnol would signicantly affect the solbility. How did the author follwed this.
3- The manuscript states: "The solubility data obtained from our studies deviate from the ones reported in the literature (see Figure )." This line indicates that there are differences between the solubility values measured in the current study and those found in existing literature, suggesting a need for further exploration of these discrepancies [1]. So to strengthen the discussion, it would be beneficial to include a comparative analysis of these values, as the manuscript does not currently elaborate on the reasons for these differences.
Reviewer 4 Report
Comments and Suggestions for Authors
This paper focus on MSZW, solubility, and their dependence on cooling rates is defined, and the application of advanced analytical techniques enhances the scientific appeal. Some minor comments are as follows:
1. There are many abbreviated terms like MSZW, FBRM, and FTIR, the passage is densely packed with jargon. Please simplify these concepts for better readability.
2. The cooling rates are main focus of this study. What are the specific values or ranges used?
3. The flow of experimental methods, mathematical methodologies, and model validation is abrupt. The organization of this manuscript should be improved.
4. “Both the nucleation rate constant and nucleation rate were found to range between 10²¹ to 10²² molecules/m³·s” should be corrected to "range between 10²¹ and 10²² molecules/m³·s”.
5. "Detailed experimental procedures were presented" could be improved for consistency. Consider rephrasing as "We presented detailed experimental procedures..."
6. The language should be improved.
Round 2
Reviewer 2 Report
Comments and Suggestions for Authors
The issues raised during the last review process were not satisfactorily addressed.
Comments on the Quality of English LanguageThe English could be improved to more clearly express the research.
Author Response
We sincerely appreciate your feedback and carefully addressed all the comments in our revised manuscript. However, we are unsure which specific comments were not fully addressed, as this was not explicitly indicated in the review.
Regarding the concern about the manuscript structure, no specific suggestions were provided on how it should be reorganized. We believe that the current structure effectively conveys the scientific content in a clear and logical manner. Furthermore, Reviewer 1 explicitly acknowledged the quality of the manuscript’s structure and raised no concerns, suggesting that the organization is accessible to readers.
For the four remaining queries related to obtaining API concentration from intensity measurements, we had already provided a detailed explanation in the revised manuscript. Additionally, we incorporated the characteristic IR peak corresponding to the API as a figure insert in Figure 1a, where we also illustrated the baseline-to-peak height measurement.
We further emphasized the novelty of our work, particularly the newly proposed mathematical expression (Equation 46), which is the first of its kind to predict nucleation rate at different cooling rates using experimentally obtained MSZW data.
Additionally, we addressed the effect of temperature on IR intensity and clarified how we accounted for and subtracted this effect in Section 3.1. In Figure 2a, we explicitly presented IR intensity measurements at different temperatures, along with the corrected values obtained after removing the temperature-induced intensity variations.
Regarding the query on fitting results, given that we obtained MSZW data at only two cooling rates, any theoretical fit would simply result in a linear line with R² = 1. Had we acquired MSZW data at three or more cooling rates, we would have been able to present a more comprehensive fit.
We appreciate your time and consideration and hope our clarifications adequately address the concerns raised.